# A small secreted protein NICOL regulates lumicrine-mediated sperm maturation and male fertility

Daiji Kiyozumi [1,2,9] ✉, Kentaro Shimada[1,3,9], Michael Chalick[4], Chihiro Emori [1], Mayo Kodani[1,3], Seiya Oura[1,3], Taichi Noda [1], Tsutomu Endo [1], Martin M. Matzuk [5], Daniel H. Wreschner [4] ✉ & Masahito Ikawa [1,3,6,7,8] ✉

The mammalian spermatozoa produced in the testis require functional maturation in the epididymis for their full competence. Epididymal sperm maturation is regulated by lumicrine signalling pathways in which testis-derived secreted signals relocate to the epididymis lumen and promote functional differentiation. However, the detailed mechanisms of lumicrine regulation are unclear. Herein, we demonstrate that a small secreted protein, NELL2-interacting cofactor for lumicrine signalling (NICOL), plays a crucial role in lumicrine signalling in mice. NICOL is expressed in male reproductive organs, including the testis, and forms a complex with the testis-secreted protein NELL2, which is transported transluminally from the testis to the epididymis. Males lacking *Nicol* are sterile due to impaired NELL2-mediated lumicrine signalling, leading to defective epididymal differentiation and deficient sperm maturation but can be restored by NICOL expression in testicular germ cells. Our results demonstrate how lumicrine signalling regulates epididymal function for successful sperm maturation and male fertility.

It is one of the prerequisites for successful sexual reproduction to produce functional spermatozoa. Spermatogenesis is promoted in the testes during puberty under the control of gonadotrophs such as luteinizing hormone and follicle-stimulating hormone[1]. Spermatogonial stem cells located basally inside the testicular seminiferous tubules divide and produce spermatocytes. Spermatocytes move toward the adluminal compartment of the seminiferous tubule by crossing the blood-testis barrier, a tight junction between Sertoli cells[2], and produce spermatids by meiosis. Spermatids then develop head and tail structures and finally differentiate into spermatozoa[3]. Once the spermatozoa are generated, they are released from the seminiferous tubule epithelium to be transported toward the epididymis. The

testicular spermatozoa formed in this manner appear morphologically complete, but they require further functional maturation in the epididymis to obtain full reproductive competence.

The epididymis is a highly coiled epithelial duct and constitutes a sperm transport route; spermatozoa coming from the testis via the efferent duct are transported through the epididymis to go out toward the vas deferens. In the epididymis, the spermatozoa become functionally mature and fully fertile under the influence of the epididymal luminal environment[4,5]. In rodents, the initial segment (IS), the most proximal region of the epididymis, is characteristic of highly differentiated tall but pseudostratified epithelial cells. When the efferent duct was ligated to interfere luminal connection between the testis and

[1]Research Institute for Microbial Diseases, Osaka University, Suita, Osaka 5650871, Japan. [2]PRESTO, Japan Science and Technology Agency, Kawaguchi, Saitama 3320012, Japan. [3]Graduate School of Pharmaceutical Sciences, Osaka University, Suita, Osaka 5650871, Japan. [4]Shmunis School for Biomedicine and Cancer Research, Tel Aviv University, Ramat Aviv 69978, Israel. [5]Center for Drug Discovery and Department of Pathology & Immunology, Baylor College of Medicine, Houston, TX 77030, USA. [6]Graduate School of Medicine, Osaka University, Suita, Osaka 5650871, Japan. [7]The Institute of Medical Science, The University of Tokyo, Minato-ku, Tokyo 1088639, Japan. [8]CREST, Japan Science and Technology Agency, Kawaguchi, Saitama 3320012, Japan. [9]These authors contributed equally: Daiji Kiyozumi, Kentaro Shimada. ✉e-mail: kiyozumi@biken.osaka-u.ac.jp; dwreschner@gmail.com; ikawa@biken.osaka-u.ac.jp

epididymis, the IS epithelium becomes degenerated[6,7]. In mouse models with impaired IS differentiation, the epididymal spermatozoa are unable to fertilize an egg[8]. It had therefore been postulated that factors synthesized in the testis are released into the lumen of seminiferous tubules and go through luminal space via an efferent duct to influence the development and function of the IS epithelial cells. This transluminal secretion was named "lumicrine" (lumi+crine) as secreted factors act via the luminal space of the male reproductive tract[9].

The molecular mechanisms of lumicrine and lumicrine-regulated sperm maturation processes in the epididymis had been uncertain until neural epidermal growth factor–like like 2 (NELL2) was identified as the first known molecular entity of lumicrine factors[10]. NELL2 is an extracellular matrix-like large protein secreted from spermatocytes in the testicular seminiferous tubule and transported transluminally to the epididymis, where NELL2 binds its cell surface receptor tyrosine kinase ROS1[6,7,9,10]. ROS1 has been known as a proto-oncogene because its constitutive kinase-active fusion product often causes non-small-cell lung cancer[11]. In mice, the Ros1 gene is expressed in the IS of the epididymis and Ros1−/− males exhibit IS differentiation failures, similar to the consequence of testicular efferent duct ligation and are completely infertile because the ejaculated spermatozoa are unable to migrate from the uterus into the oviduct in the female reproductive tract[8,10,12]. Upon endogenous stimulation by NELL2, ROS1 activates the intracellular signalling pathway and triggers epididymal epithelial differentiation. The fully differentiated epididymal epithelium expresses many proteins that define the epididymal luminal environment necessary for sperm maturation. Thus, NELL2 and ROS1 define the axis of testis-epididymis lumicrine signalling. The detailed molecular mechanism of lumicrine-mediated sperm maturation, however, remains largely unclear because there are more molecular components other than NELL2 and ROS1 expected to be included but remained unidentified.

Here we show that the secreted protein NICOL constitutes the lumicrine signalling pathway. Nicol (a predicted gene Gm1673)-null males are infertile because of deficient epididymal differentiation and subsequent sperm maturation and phenocopies lumicrine-deficient animals such as Nell2−/− or Ros1−/− males. NICOL forms a tight molecular complex with NELL2 to transmit a lumicrine signal which is indispensable for sperm maturation and male fertility. Identification of NICOL will promote the potential development of non-hormonal male contraceptives that target lumicrine signalling pathways.

## Results

### NICOL is required for male fertility
Nicol, a mouse ortholog of human C4orf48, encodes a small secreted protein of unknown function (Fig. 1a)[13,14]. Nicol expression is enriched in both male and female reproductive organs, including the testis, epididymis, seminal vesicles, coagulating glands, ovary, and uterus, and in various non-reproductive organs in mice, as revealed by reverse transcription-polymerase chain reaction (RT-PCR) analyses (Fig. 1b). We examined the physiological functions of Nicol in mice using CRISPR/Cas9-mediated genome editing and a Nicol-null allele, i.e., Nicolem1 (1862 bp deletion, hereafter Nicol−), which lacks the whole protein coding sequence (Supplementary Fig. 1). Nicol−/− homozygous males are sterile, whereas Nicol+/− heterozygous males and females and Nicol−/− homozygous females are fertile (Fig. 1c). Apart from male fertility, Nicol seems dispensable as the gross appearance, behaviours such as mating and feeding, and the growth were not critically affected in Nicol−/− males (Supplementary Fig. 1). Nicol−/− male infertility was rescued by systemic expression of Nicol using a Nicol transgene driven by the CAG promoter (Fig. 1d). Although Nicol was abundantly expressed in the brain and testis, Nicol ablation did neither affect the gross appearance of the testis nor spermatogenesis within the seminiferous tubules (Fig. 1e–k), excluding dysfunction of the

hypothalamic–pituitary–gonadal axis[15]. These results indicate that Nicol is indispensable for male fertility.

### Nicol-null spermatozoa are unable to migrate in the female reproductive tract
The successful formation of copulatory plugs in female mice mated with Nicol−/− males (Fig. 1c, d) excluded the possibility that the observed infertility was a consequence of neither a systemic abnormality nor mating behaviour defects. We then investigated the migration of spermatozoa from Nicol−/− males ejaculated into the female reproductive tract by visualizing the spermatozoa using fluorescent protein expression (Fig. 2a)[16]. Spermatozoa ejaculated by Nicol+/− males successfully migrated from the uterus into the oviduct (Fig. 2b–d), while sperm from Nicol−/− males remained within the uterus and did not migrate beyond the utero-tubal junction (Fig. 2e–g). In addition to their defective migration in vivo, Nicol-null spermatozoa isolated from the cauda epididymis were unable to bind to the egg zona pellucida (ZP) in vitro (Fig. 2h–j). The fertilization of cumulus-free eggs in vitro was thus dramatically reduced (Fig. 2k), whereas the fertilization of cumulus-intact and ZP-free oocytes were not critically affected (Fig. 2l, m). The cumulus-intact oocytes fertilized with Nicol-null spermatozoa successfully developed into blastocysts when cultured in vitro (Fig. 2n–p).

### Defective sperm maturation in the epididymis in Nicol−/− males
Defective sperm migration from the uterus into the oviduct and poor sperm binding to the ZP are known to result from loss of the mature form of the sperm surface transmembrane protein a disintegrin and metallopeptidase 3 (ADAM3)[17]. ADAM3 is expressed as a 100 kDa precursor in testicular germ cells and is then processed into a mature form of ~25 kDa by limited proteolysis when the spermatozoa transit the epididymis (Fig. 3a)[10]. Expression levels of mature processed ADAM3 were low or absent in cauda epididymal spermatozoa from Nicol−/− mice (Fig. 3b) because of its aberrant processing during sperm transit through the epididymis (Supplementary Fig. 2), whereas expression levels of ADAM3 precursor and germ cell-intrinsic factors necessary for ADAM3 expression, including ADAM2, calmegin, calsperin, protein disulfide-isomerase-like protein of the testis (PDILT), and testicular angiotensin converting enzyme (tACE)[18–22] were not compromised (Fig. 3c). These results indicate that epididymal sperm maturation was defective as a consequence of Nicol deletion.

The secreted proteases ovochymase 2 (OVCH2) and ADAM28 are abundantly expressed in caput epididymis[10,23] and potentially act on the sperm surface. Males lacking OVCH2 are unable to process sperm ADAM3 and are sterile because of defective sperm migration into the oviduct[10,24]. Expression levels of OVCH2 and ADAM28 were significantly diminished in the IS-caput epididymis in Nicol−/− mice compared with wild-type (WT) mice (Fig. 3d), indicating that aberrant ADAM3 processing in sperm in Nicol−/− mice is a consequence of attenuated OVCH2 expression.

### Epithelial differentiation is ablated in Nicol−/− epididymis
OVCH2 is induced upon differentiation of the epididymal initial segment (IS) epithelium in a lumicrine-dependent manner[10]. The luminal epithelium differentiated postnatally and thickened in WT IS but remained thin and did not differentiate in Nicol−/− mice (Fig. 4a–d and Supplementary Fig. 3). The degenerated IS was apparently a phenocopy of Nell2−/− (Fig. 4e), Ros1−/− (Fig. 4f), efferent duct-ligated (Fig. 4g), or KitW/W-v (W/Wv, Fig. 4h) mice, which show impaired lumicrine signalling[6,10,12]. No apparent histological abnormalities were observed in the corpus and cauda epididymis in Nicol−/− mice (Supplementary Fig. 4).

Differentiation of the IS epithelium is regulated by NELL2–ROS1-mediated lumicrine signalling[10]. ROS1 autophosphorylation activates

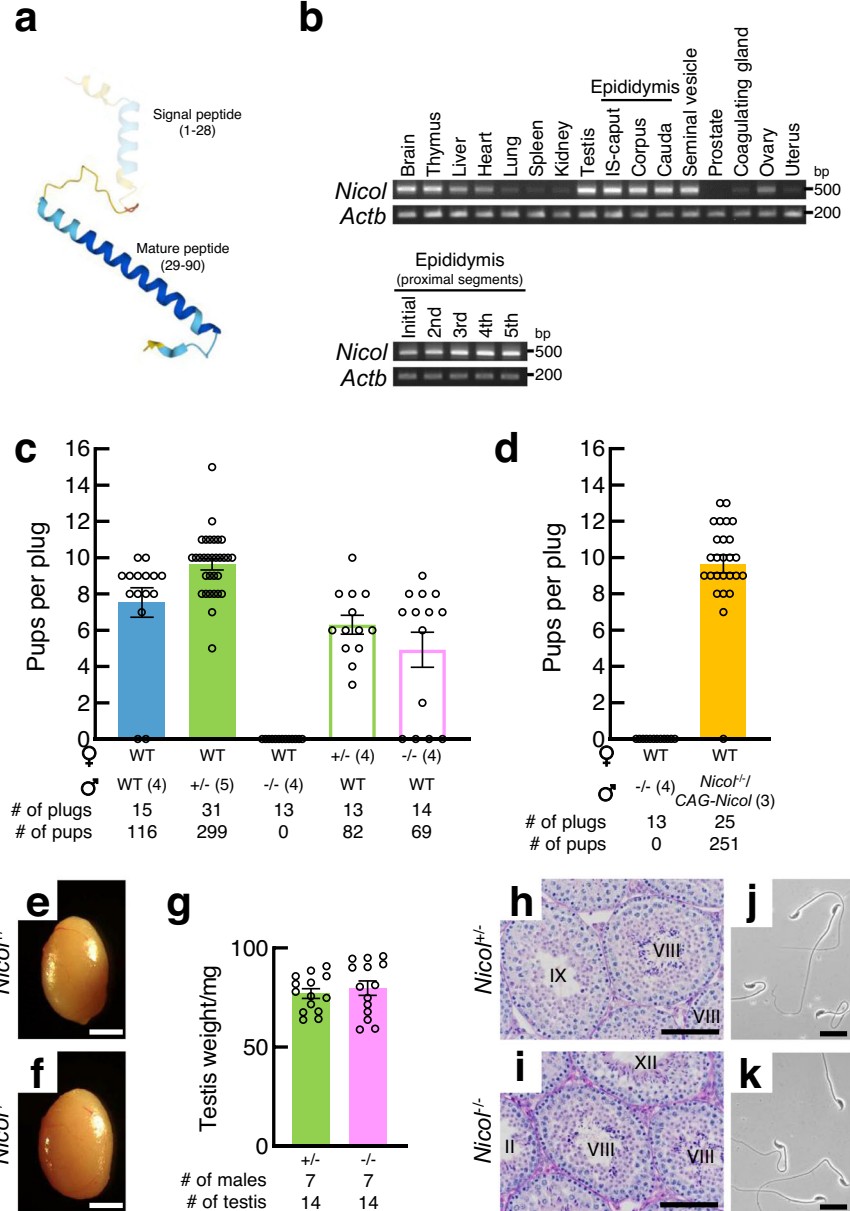

**Fig. 1 | Nicol encodes a secreted protein that is indispensable for male fertility.**
**a** AlphaFold-predicted 3D structure of mouse NICOL. The signal peptide is depicted as faded. **b** RT-PCR analyses of Nicol expression in adult mouse organs, with Actb as an internal control. The initial to the fifth proximal segments of IS-caput region of epididymis was also separately analysed. Images are representative ones obtained from three independent biological replicates. **c** Litter sizes of WT, Nicol[+/−], and Nicol[−/−] male (filled columns) and Nicol[+/−] and Nicol[−/−] female mice (unfilled columns). Average and standard error (S.E.) shown. **d** Litter sizes of Nicol[−/−] and Nicol[−/−]/CAG-Nicol males. Average and S.E. shown. **e–k** Appearance of testis (**e, f**), testicular weight (**g**), HE staining of testis sections (**h, i**), and epididymal sperm morphologies (**j, k**) in Nicol[+/−] (**e, h, j**) and Nicol[−/−] (**f, i, k**) mice. Stages of spermatogenesis indicated by Roman numerals (**h, i**). Images are representative ones obtained from three independent biological replicates (**e, f, h–k**). For bar plots, values are shown as mean ± S.E.M. P value = 0.5538 was determined by a two-tailed unpaired Students' t-test (**g**). Scale bars, 2 mm (**e, f**), 100 μm (**h, i**), 20 μm (**j, k**).

the extracellular signal-regulated kinase (ERK) signalling pathway[25]. ERK1/2 phosphorylation levels in the IS-caput epididymis were lower in Nicol[−/−] compared with control heterozygous mice (Fig. 4i), similar to Nell2[−/−], Ros1[−/−], and efferent duct-ligated mice[10,12,26]. RNA-seq analyses clearly showed that most downregulated genes in the IS-caput epididymis were common to Nicol[−/−] and Ros1[−/−] mice (Supplementary Fig. 5). In addition, expression levels of Etv1, Etv4, and Etv5, which are located downstream of the ERK signalling pathway, were also significantly decreased in the Nicol[−/−] IS-caput epididymis (Fig. 4j). As observed in the spermatozoa of Ros1[−/−] mice[8], the flagella of Nicol-null spermatozoa became bent during epididymal transit and the sperm motility

characterized by computer-assisted sperm analysis (CASA) altered slightly (Supplementary Fig. 6), endorsing the abnormality of sperm maturing function of Nicol[−/−] IS epididymis. Collectively, these results show that differentiation of the epididymal IS is defective in Nicol[−/−] mice.

During lumicrine signalling, NELL2 binds to the extracellular region of ROS1, while SHP-1, a tyrosine phosphatase encoded by Ptpn6, binds to the ROS1 intracellular region to allow ROS1-mediated signalling[10,12,27]. Gene expression levels of Nell2, Ros1, and Ptpn6 in the IS-caput epididymis and testis were not compromised in Nicol[−/−] mice (Fig. 4k), implying a direct action of NICOL in the lumicrine signalling pathway.

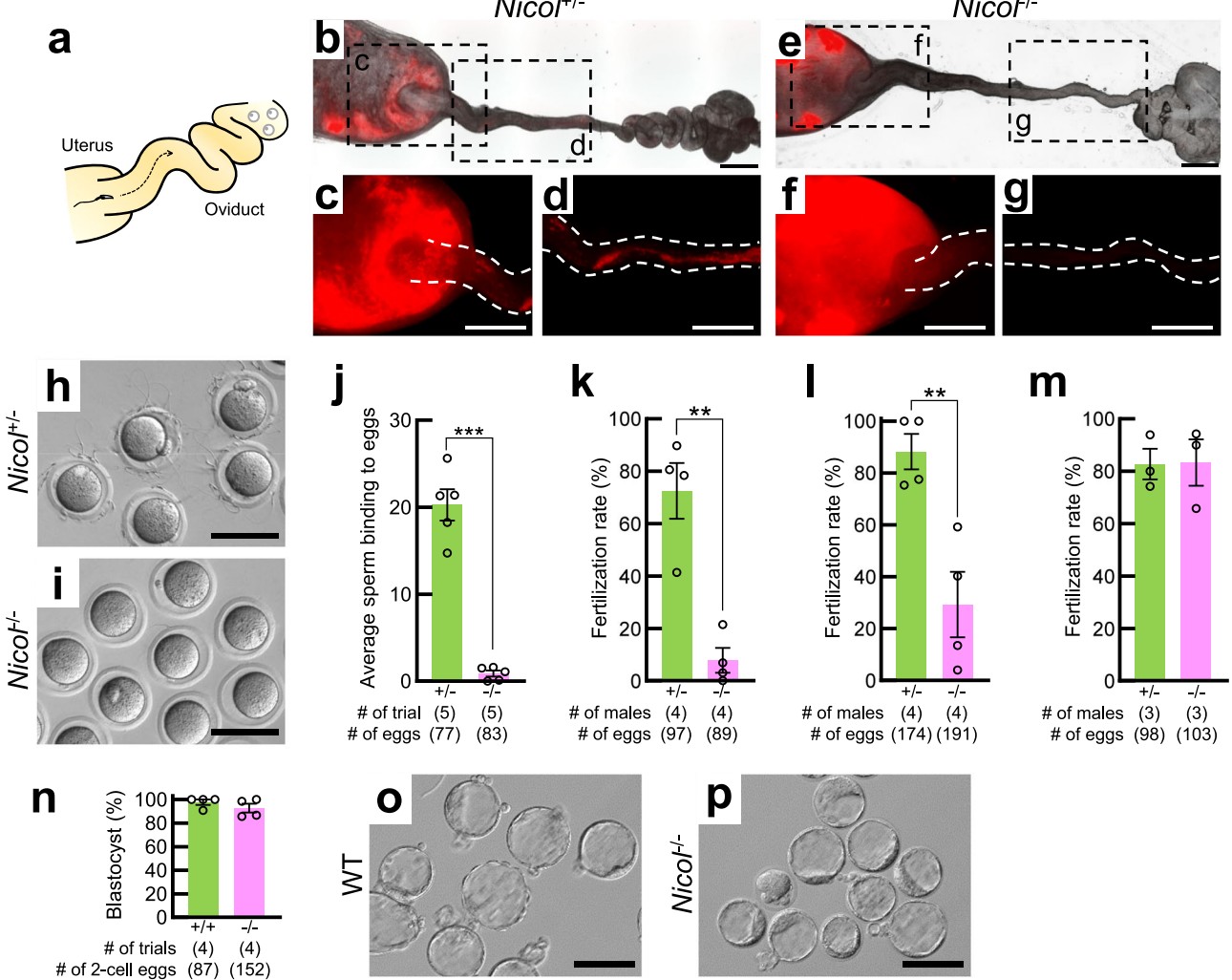

**Fig. 2 | Defective sperm maturation blocks sperm migration into the oviduct resulting in male sterility in *Nicol⁻/⁻* mice. a** Schematic representation of ejaculated sperm migrating from the uterus into the oviduct towards oocytes. **b–g** Migration of red fluorescence-illuminated sperm from *Nicol⁺/⁻* (**b–d**) and *Nicol⁻/⁻* (**e–g**) mice ejaculated into reproductive tract in WT females. Bars, 1 mm. **h, i** Representative images of sperm–ZP binding assay using *Nicol⁺/⁻* (**h**) and *Nicol⁻/⁻* (**i**) sperm obtained from five independent biological replicates. Bars, 100 μm. **j** Sperm–ZP binding assay using sperm from *Nicol⁺/⁻* (green columns) and *Nicol⁻/⁻* (pink columns) mice. Average and S.E. shown. **k–m** In vitro fertilization capacity of sperm from *Nicol⁺/⁻* and *Nicol⁻/⁻* mice in assays with cumulus-free (**k**), cumulus-intact (**l**), and ZP-free (**m**) oocytes. **n** Development of fertilized eggs into blastocysts. **o, p** Representative images of blastocysts derived from WT oocytes inseminated with spermatozoa of WT (**o**) or *Nicol⁻/⁻* (**p**) males obtained from four independent biological replicates. Bars, 100 μm. For bar plots, values are shown as mean ± S.E.M. *P* value = 0.00001 (**j**), 0.0015 (**k**), 0.0075 (**l**), 0.9565 (**m**), and 0.3052 (**n**) were determined by a two-tailed unpaired Students' *t*-test. **P* < 0.05, ***P* < 0.01, ****P* < 0.001.

## NICOL is a cofactor of NELL2

*Nicol* expression in the testis increases postnatally (Fig. 5a)[28], and *Nicol* is expressed in both germ cells and Sertoli cells (Fig. 5b). Further analysis indicates that both *Nicol* and *Nell2* are strongly expressed in the spermatocyte subpopulation (Fig. 5b, c), and *Nicol* is expressed in *Nell2*-positive cells (Fig. 5d). Because of the presence of the blood–testis barrier formed by tight junctions between Sertoli cells (Fig. 5e), secretions from pachytene spermatocytes constitute the testicular luminal fluid, which flows via efferent ducts towards the epididymis. We investigated the molecular functions of NICOL in the luminal fluid by expressing recombinant NICOL protein in mammalian cells and purifying it from the conditioned medium (Supplementary Fig. 7). Purified recombinant NICOL protein conjugated to agarose beads specifically pulled down NELL2 (Fig. 5f). To confirm these findings in vivo, *Nicol* was expressed in a testicular germ cell-specific manner by the *Clgn* promoter-driven *Nicol* transgene[29]. NELL2 was co-immunoprecipitated with NICOL from testis lysates of mice carrying the *Clgn-Nicol* and *Clgn-Nell2* transgenes, which enable testicular NICOL and NELL2 proteins to be immunodetected (Fig. 5g), confirming the potential of NICOL to function in a complex with NELL2 in vivo. The association rate constant $k_a$, dissociation rate constant $k_d$, and dissociation equilibrium constant $K_D$ for the interaction between purified recombinant NELL2 and NICOL proteins determined by surface plasmon resonance technology were $7.2 \times 10^4\,M^{-1}s^{-1}$, $6.2 \times 10^{-3}\,s^{-1}$, and 87 nM (Fig. 5h), respectively, indicating that NELL2 and NICOL constituted a tight heteromeric complex. In addition, NICOL-conjugated beads pulled down the ROS1 ectodomain, indicating that NICOL also bound to ROS1 (Fig. 5i). The NELL2–NICOL complex might thus be multivalent for ROS1 binding. These results suggest that NICOL modulates NELL2–ROS1-mediated lumicrine signalling by acting directly on NELL2.

## NICOL is a component of lumicrine signalling

*Nicol* is expressed in the epididymis and *Nicol* ablation caused defective epididymal IS differentiation; however, the above in vivo and in vitro observations suggest that NICOL expressed in the testis acts

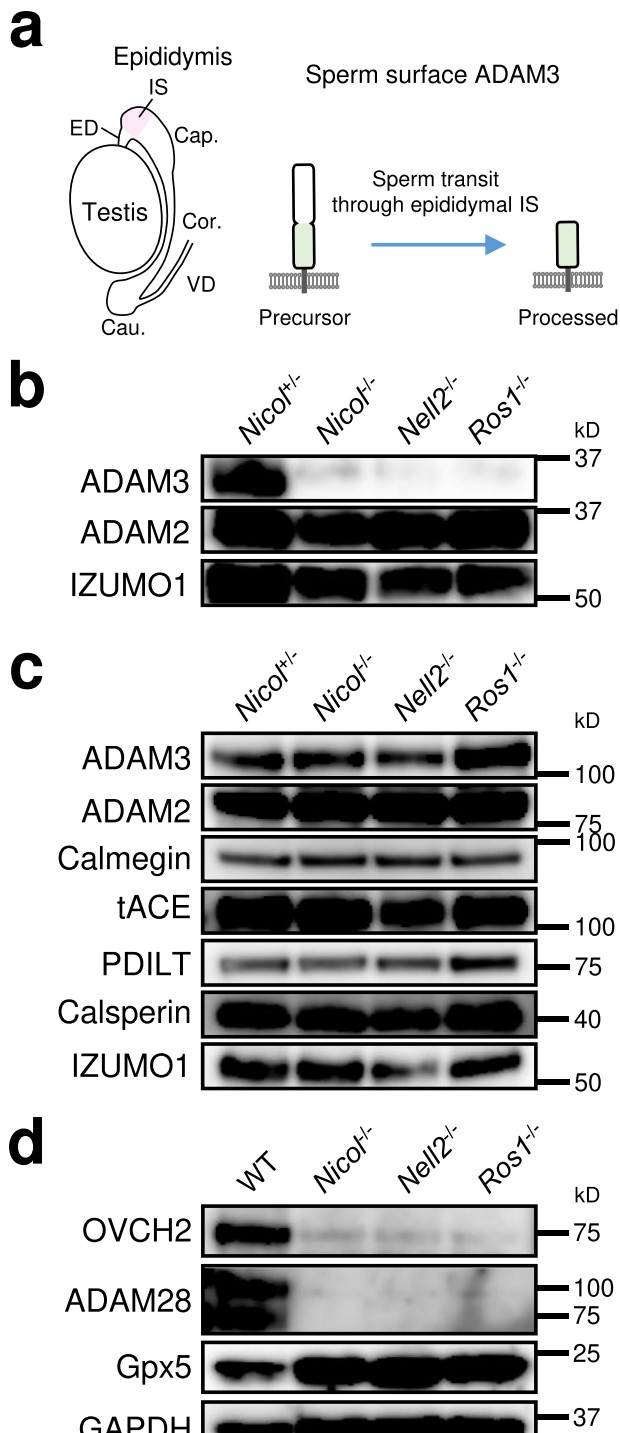

**Fig. 3 | Defective epididymal sperm maturation in *Nicol⁻/⁻* male mice. a** A schematic representation of ADAM3 processing during sperm transit through epididymal IS. **b** Expression of proteins associated with sperm ability to migrate into the oviduct and ZP binding of cauda epididymal sperm from *Nicol⁺/⁻* and *Nicol⁻/⁻* mice. **c** Expression of proteins associated with sperm ability to migrate into the oviduct and ZP binding in testicular germ cells from *Nicol⁺/⁻* and *Nicol⁻/⁻* mice. **d** Expression of OVCH2 and ADAM28 in WT, *Nicol⁻/⁻*, *Nell2⁻/⁻*, and *Ros1⁻/⁻* IS-caput epididymis. Gpx5 and GAPDH shown as internal controls. Images are representative ones obtained from three independent biological replicates (**b–d**).

together with NELL2 to secure testis–epididymis lumicrine signalling. We examined the action of NICOL in lumicrine signalling by restricting *Nicol* expression only in the testis (Fig. 6a, b). Introduction of the *Clgn-Nicol* transgene into a *Nicol⁻/⁻* genetic background successfully rescued

IS differentiation and accordingly increased the height of the IS epithelium (Fig. 6c, d). Phosphorylation of ERK and expression of ETV5, OVCH2, and ADAM28, which are induced during epididymal IS differentiation[10], were also restored (Fig. 6e), and the processing of ADAM3 to a mature form was also restored in cauda spermatozoa of *Nicol⁻/⁻/Clgn-Nicol* mice (Fig. 6f). Infertility of *Nicol⁻/⁻* male mice was accordingly completely rescued by testicular germ cell-specific NICOL expression using the *Clgn-Nicol* transgene (Fig. 6g). Collectively, these results indicate that NICOL functions in vivo as a component of lumicrine signalling and plays a crucial role in epididymal-mediated sperm maturation.

## Discussion

Epididymal sperm maturation is essential for their fertilization ability; however, the molecular mechanisms mediating this maturation are poorly understood compared with the mechanisms regulating testicular spermatogenesis. Lumicrine signalling in the male reproductive tract has been shown to function as a master regulator of sperm maturation and its molecular mechanisms have been partially elucidated[10,24]. Although various factors are predicted to play roles in the series of lumicrine regulatory mechanisms, i.e., expression of secreted factors in the testis, trans-luminal transport, reception of testicular secreted factors in the epididymis, and activation of the epididymal sperm maturation machinery, few of the responsible factors and their functions have been identified to date. Sperm defects or dysfunction are the most common causes of infertility[30], and elucidation of the pathways participating in the lumicrine regulation of sperm maturation is therefore essential to clarify the causes of male infertility. The current findings revealed the involvement of the small secreted protein NICOL in lumicrine signalling, and showed that NICOL functions together with NELL2 to constitute a testis-derived lumicrine factor that triggers epididymal differentiation.

Based on the present study, a current model of lumicrine signalling system is suggested (Fig. 7). The testicular seminiferous luminal fluid is generated by the function of ion transporters and aquaporins in Sertoli cells[31] and NICOL is secreted by exocytosis into seminiferous luminal fluid where it can form a complex together with NELL2 (Fig. 7a). NICOL and NELL2 are transported from testis toward epididymis by the luminal fluid flow (Fig. 7b). Although spermatozoa are suggested to be as the carrier of lumicrine factor molecules[32], they seem dispensable for lumicrine signalling as the IS differentiation starts before the completion of the first wave of spermatogenesis (Supplementary Fig. 3)[33]. On the apical surface of IS epithelial cells, NICOL-NELL2 complex bind and multimerize ROS1 to trigger IS differentiation which is necessary for sperm maturation (Fig. 7c).

There is a growing social need for male contraceptives. However, the development of hormone-based male contraceptives is undesirable because of the potential side effects associated with the modulation of testosterone production, which acts to maintain sexual function as well as bone and muscle mass[34]. There are currently no non-hormonal male contraceptives on the market, and efforts are therefore focused on identifying sperm-specific proteins and pathways that could serve as drug targets for the development of safe and effective male contraceptives. Targeting sperm production by depleting spermatogenic cells or interrupting spermatogenesis seems less suitable because it alters the cellular composition and microenvironment in the testis, which may in turn trigger the hypothalamus–pituitary–testis feedback system and cause toxic side effects. Lumicrine signalling-mediated sperm maturation could be an alternative target for male contraceptive pills, and inhibition of ROS1 kinase activation has been identified as a possible target for male contraception. Although attempts to use crizotinib, a tyrosine kinase inhibitor developed against c-Met but also effective against ALK and ROS1 tyrosine kinases[35–37], for male contraception were unsuccessful[38], the identification of NICOL as a component of NELL2 and

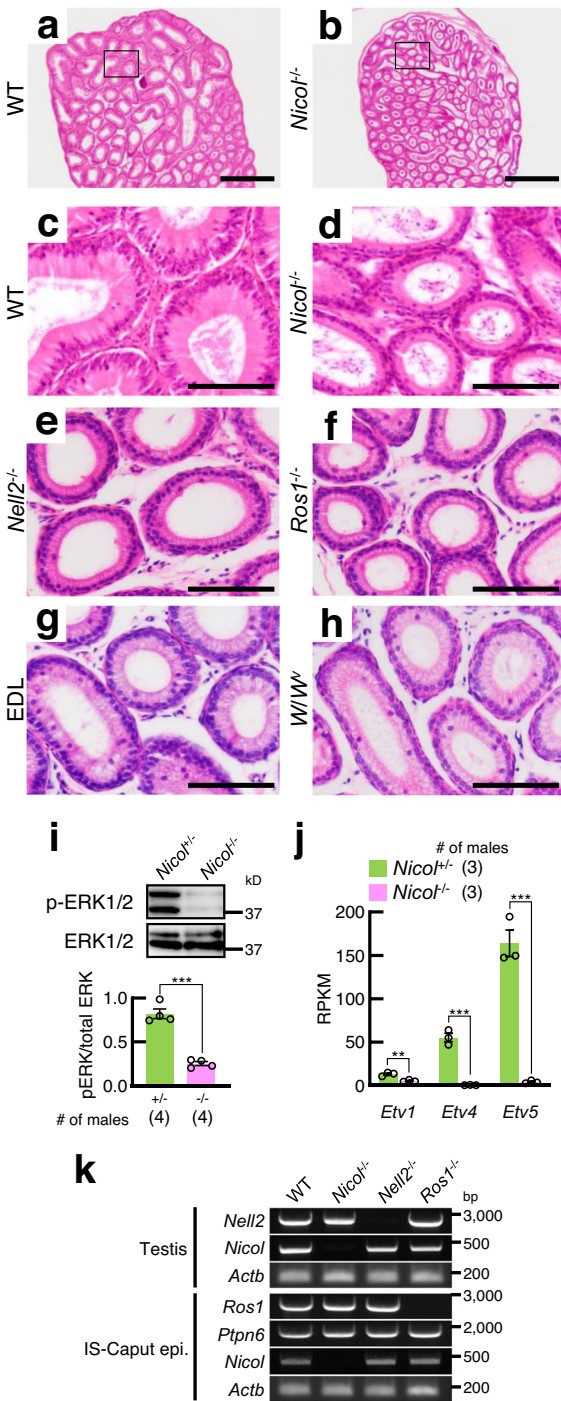

**Fig. 4 | Defective epididymal IS epithelium differentiation in *Nicol*⁻/⁻ mice is a phenocopy of other lumicrine signalling-deficient mice.** a–h HE staining of IS sections of epididymis from WT (**a, c**), *Nicol*⁻/⁻ (**b, d**), *Nell2*⁻/⁻ (**e**), *Ros1*⁻/⁻ (**f**), efferent duct-ligated (EDL) (**g**), and *W/Wᵛ* (**h**) mice. Bars, 500 μm (**a, b**), 100 μm (**c–h**). Images are representative ones obtained from three (**a–d, g, h**) or five (**e, f**) independent biological replicates. **i** Immunoblot detection of phosphorylated and total ERK1/2 in *Nicol*⁺/⁻ (green columns) and *Nicol*⁻/⁻ (pink columns) IS-caput epididymis. Average ERK phosphorylation levels and S.E. shown. **j** RNA expression of ERK downstream transcription factor genes *Etv1*, *Etv4*, and *Etv5* in *Nicol*⁺/⁻ and *Nicol*⁻/⁻ IS-caput epididymis. Average and S.E. shown. RPKM, reads per kilobase per million. **k** RT-PCR analyses of *Nell2*, *Nicol*, *Ros1*, and *Ptpn6* expression in *Nicol*⁻/⁻ testis and caput epididymis. *Actb* shown as internal control. Images are representative ones obtained from three independent biological replicates. For bar plots, values are shown as mean ± S.E.M. *P* value = 0.0001 (**i**) and 0.006, 0.0004, and 0.0005 for *Etv1*, *Etv4*, and *Etv5*, respectively (**j**), were determined by a two-tailed unpaired Students' *t*-test. *P < 0.05, **P < 0.01, ***P < 0.001.

ROS1-mediated lumicrine signalling reported here would expand the options to develop male contraceptives with fewer off-target effects, based on a molecular basis including eliminating interactions among NICOL, NELL2, and ROS1.

## Methods

### Animals

B6D2F1 and *W/Wᵛ* mice were purchased from Japan SLC, Inc. *Nell2* knockout (KO) mice, *Ros1* KO mice, *Clgn-Nell2* transgenic mice, *Adam3* KO mice, and red body green sperm (RBGS) transgenic mice were obtained previously[10,16,39]. For efferent duct ligation, the efferent ducts of 10-week-old B6D2F1 males were unilaterally ligated and the ipsilateral epididymis was excised 4 weeks after ligation. *Nicol* KO mice (B6D2-Gm1673 <em1Osb>) were generated on a B6D2F1 background using CRISPR/Cas9-mediated genome editing. Briefly, crispr (cr) RNA#1 and crRNA#2 (Sigma, custom synthesis), SygRNA SpCas9 tracrRNA (Sigma, #TRACRRNA05N-5NMOL), and TrueCut™ Cas9 Protein v2 (ThermoFisher, #A36496) were injected into fertilized eggs. The crRNA sequences and genotyping primer sequences are listed in Supplementary Tables 1, 2, respectively. *CAG-Nicol* and *Clgn-Nicol* transgenic mice were generated by injecting a DNA fragment carrying the *CAG* or *Clgn* promoter[16], respectively, a cDNA encoding NICOL-PA-Rho1D4, and a polyadenylation signal into pronuclei of fertilized B6D2F1 eggs. The mouse lines generated in this study have been deposited as frozen sperm at the RIKEN BioResource Research Center (BRC) and Center for Animal Resources and Development (CARD) at Kumatomo University, where they will be made available to all researchers. The BRC and CARD repository IDs for the gene-modified mouse lines generated in this study are as follows: B6D2-Gm1673<em1Osb>, RBRC11231 and 3035; B6D2-Tg(Clgn-Gm1673/PA/1D4)2Osb, RBRC11479 and 3123; B6D2-Gm1673<em1Osb> Tg(CAG-Gm1673/PA/1D4)1Osb, and RBRC11486 and 3130, respectively. Because of the difficulty in dissecting IS separately from caput epididymis especially in mutant mice in which IS differentiation is ablated, the IS was dissected together with the caput and such tissue dissection was indicated by the description "IS-caput." Animals were maintained under 12 h light (8:00–20:00) and 12 h dark (20:00–8:00) cycle, ambient temperature of $21 \pm 1\,^\circ\text{C}$, and $55 \pm 10\%$ humidity. All experiments involving animals were approved by the Institutional Animal Care and Use Committees of Osaka University (Osaka, Japan) and were conducted in compliance with the university guidelines and regulations for animal experimentation.

### Mating test

Male mice were mated with 2-month-old B6D2F1 WT female mice for several months. Females were inspected for copulatory plug formation and delivery every morning.

### In vitro fertility test

Spermatozoa isolated from the cauda epididymis were dispersed in a drop of Toyoda, Yokoyama, Hoshi (TYH) medium[40] covered with paraffin oil and capacitated by incubating for 2 h at 37 °C under 5% $CO_2$. Cumulus–oocyte complexes (COCs) were collected from superovulated B6D2F1 females, placed in a TYH drop, and treated with 300 μg/ml hyaluronidase (Sigma, #H4272) for 5 min to remove the cumulus layer. The ZP was removed from cumulus-free oocytes by treating with 1 mg/ml collagenase (Sigma, #C5138) for 5 min. The capacitated spermatozoa were then added to the oocytes in the TYH drop at final concentrations of $2 \times 10^5$, $2 \times 10^5$, and $2 \times 10^4$ spermatozoa/ml for cumulus-intact, cumulus-free, and ZP-free oocytes, respectively. After 8 h of insemination, the formation of pronuclei was examined. Two-cell zygotes derived from cumulus-intact oocytes were developed into blastocysts in vitro or transplanted into pseudopregnant ICR females to obtain pups.

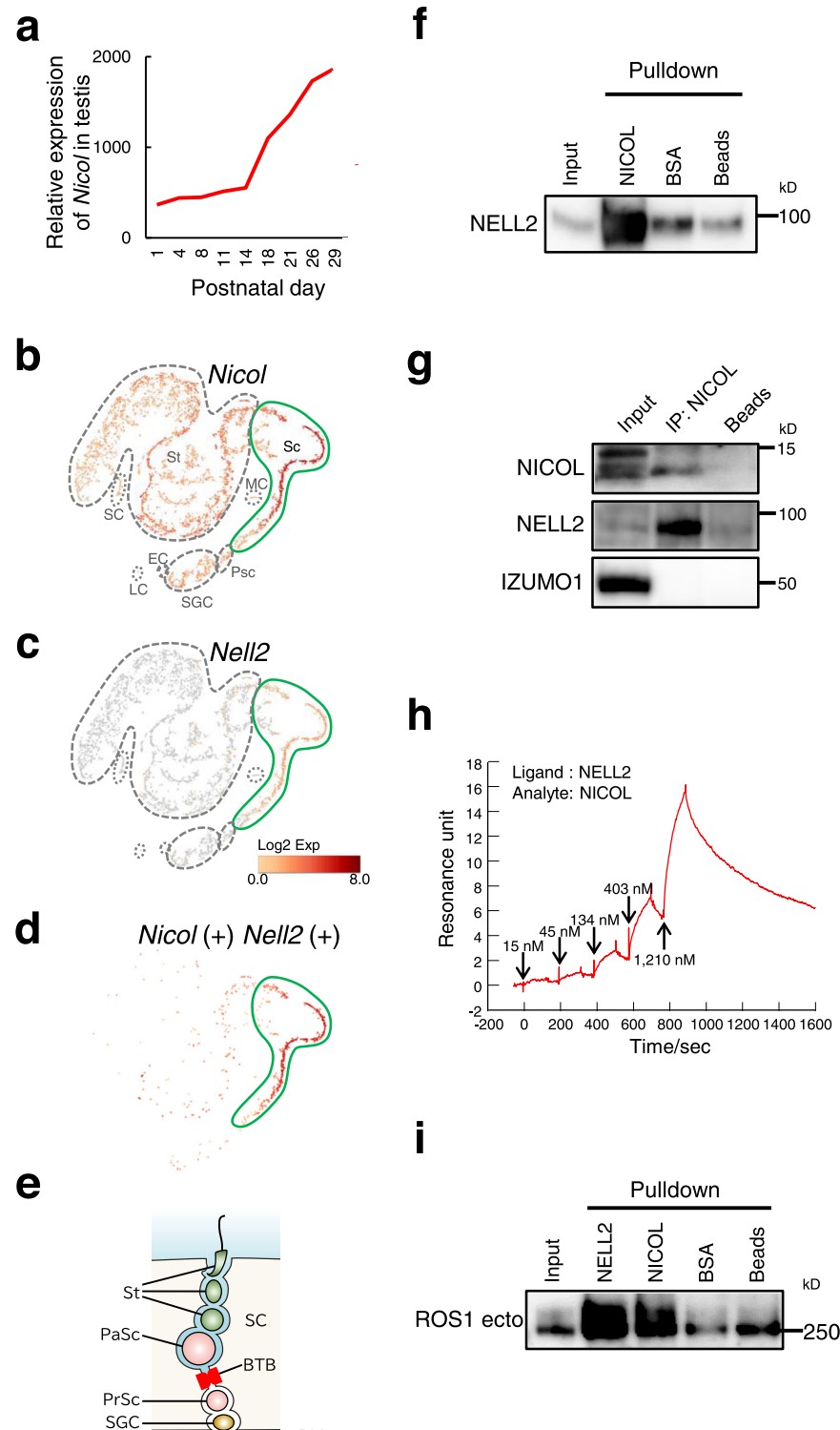

**Fig. 5 | NICOL forms a complex with NELL2. a** RNA expression profile of *Nicol* in developing testis. **b**–**d** *t*-distributed stochastic neighbour embedding plots of adult mouse testis single-cell RNA-seq data representing *Nicol* (**b**) and *Nell2* (**c**) expression. *Nicol* expression in *Nell2*-expressing cells is also shown (**d**). *Nicol* is broadly expressed with highest expression in leptotene, pachytene, and diplotene spermatocyte populations, which also show the highest *Nell2* expression. SGC spermatogonial stem cells, P Preleptotene spermatocytes, LZPD leptotene, zygotene, pachytene, and diplotene spermatocytes, St spermatids, SC Sertoli cells, EC endothelial cells, MC myoid cells, LC lymphatic cells. **e** Schematic representation of seminiferous tubule histology. SGC spermatogonial stem cell, PrSc preleptotene spermatocyte, PaSc pachytene spermatocyte, St spermatid, SC Sertoli cell, BTB

blood–testis barrier, BM basement membrane. **f** In vitro NELL2 pulldown with NICOL or BSA. Images are representative ones obtained from three independent biological replicates. **g** Co-immunoprecipitation analyses of NICOL with NELL2 from *Clgn-Nicol* and *Clgn-Nell2* testis lysate. Izumo sperm-egg fusion protein 1 (IZUMO1) shown as an internal negative control. Images are representative ones obtained from three independent biological replicates. **h** Kinetics of NELL2–NICOL interaction analysed by surface plasmon resonance technology. Binding of NICOL at indicated concentrations onto immobilized NELL2. **i** In vitro ROS1 ectodomain pulldown with NELL2, NICOL, or BSA. The image is representative one obtained from three independent biological replicates.

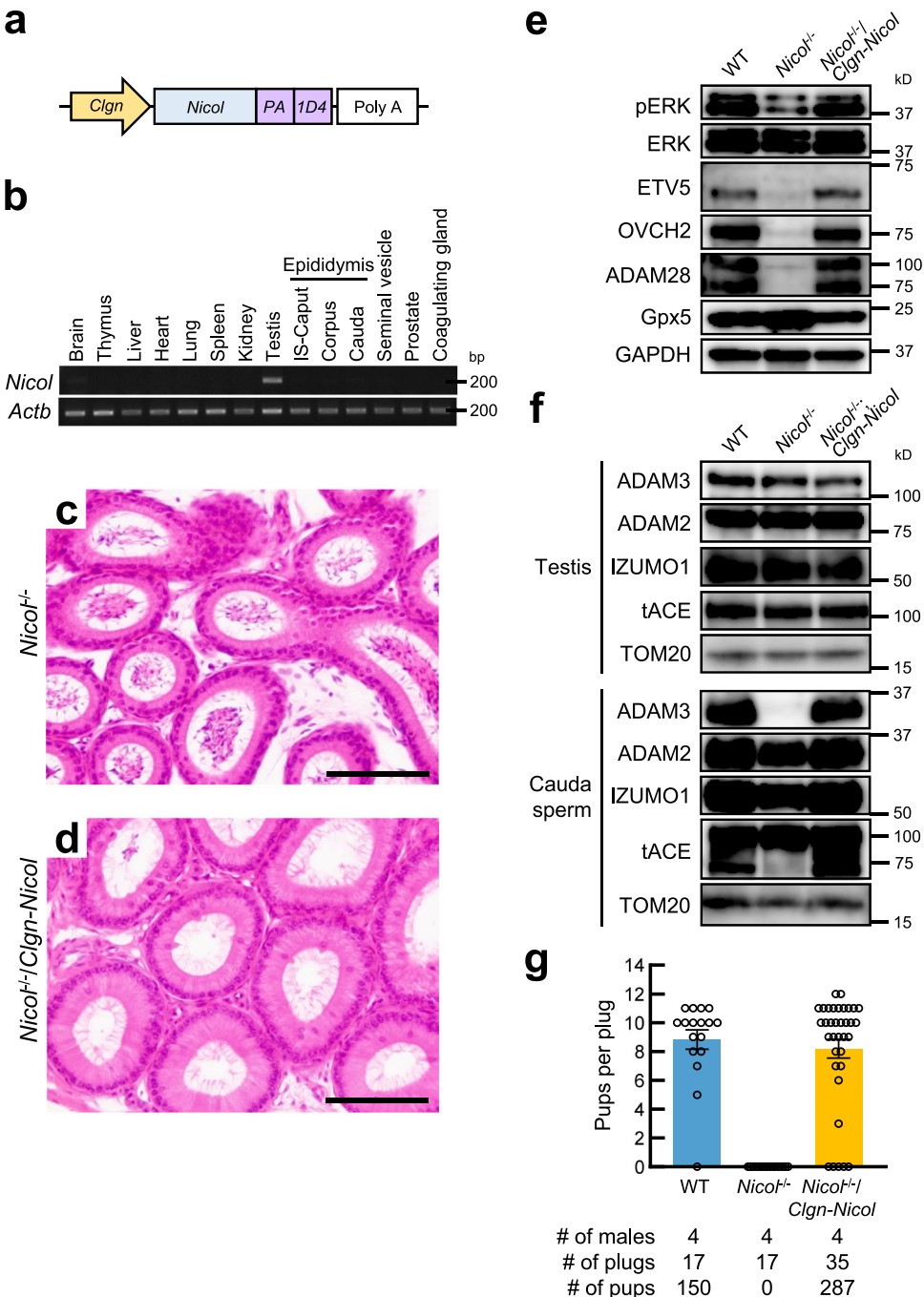

**Fig. 6 | Testicular germ cell-specific expression of *Nicol* rescued epididymal differentiation, sperm maturation, and male infertility in *Nicol*⁻/⁻ mice.**
**a** Schematic representation of transgene *Clgn-Nicol*. **b** RT-PCR analyses of *Clgn-Nicol* transgene in brain (Br), testis (Te), and caput epididymis (Cap) in *Nicol*⁻/⁻/*Clgn-Nicol* mice. *Actb* shown as internal control. Images are representative ones obtained from three independent biological replicates. **c, d** HE staining of *Nicol*⁻/⁻ (**c**) and *Nicol*⁻/⁻/*Clgn-Nicol* (**d**) IS. Bars, 100 µm. **e** Immunoblot analyses of phosphorylated ERK, ETV5, OVCH2, and ADAM28 in WT, *Nicol*⁻/⁻, and *Nicol*⁻/⁻/*Clgn-Nicol* IS-caput

epididymis. Gpx5 and glyceraldehyde 3-phosphate dehydrogenase (GAPDH) shown as internal controls. Images are representative ones obtained from three independent biological replicates. **f** Immunoblot analyses of mature ADAM3 expression in WT, *Nicol*⁻/⁻ and *Nicol*⁻/⁻/*Clgn-Nicol* cauda epididymal sperm. **g** Litter sizes of WT, *Nicol*⁻/⁻, and *Nicol*⁻/⁻/*Clgn-Nicol* males. Images are representative ones obtained from three independent biological replicates (**e**, **f**). For bar plots, values are shown as mean ± S.E.M.

## Sperm motility analysis

Spermatozoa were isolated from the cauda epididymis and suspended into a 100 µl drop of TYH medium. After 10 min or 120 min incubation, spermatozoa were obtained from the top of the drops and CASA was done using the CEROS II (software version 1.5; Hamilton Thorne Biosciences, Beverly, MA, USA) sperm analysis system.

## Antibodies

The following primary antibodies were used: mouse monoclonal anti-ADAM2 (#MAB19292; Millipore, Temecula, CA, USA), anti-ERK1/2 (#4695) and anti-phospho-ERK1/2 (#4370) (Cell Signaling Technology), anti-ADAM28 (#22234-1-AP), anti-NELL2 (#11268-1-AP), and anti-ETV5 (#13011-1-AP) (all ProteinTech), anti-PRSS37 (#HPA020541),

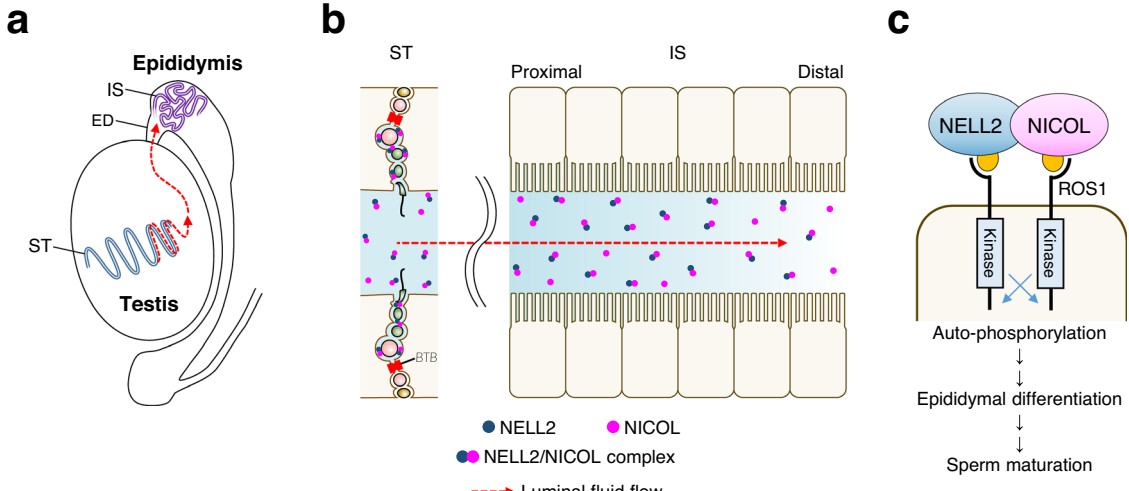

**Fig. 7 | Possible working model of NICOL in lumicrine signalling. a** The luminal fluid inside testicular seminiferous tubules flows and reaches epididymal IS via efferent duct. **b** The NELL2 and NICOL proteins secreted into seminiferous luminal fluid form a complex and reach the epididymal IS by the luminal flow. Spermatocytes seems to play a critical role in lumicrine factor secretion as they express both *Nell2* and *Nicol* at the highest level in testicular cells. A contribution to the luminal NICOL protein by testicular and epididymal cells other than germ cells is not excluded, although such a non-germ cell contribution is dispensable. See also Fig. 4e for the schematic representation of a seminiferous tubule. **c** On IS cell surface NELL2-NICOL complex binds to ROS1 leading to its multimerization and subsequent self-activation, which eventually induces IS differentiation and sperm maturation.

anti-ADAM7 (#HPA008879), and anti-FLAG (#F1804) (all Sigma), anti-ADAM3 (#sc-365288), anti-Gpx5 (#sc-390092), and anti-TOM20 (#sc-11415) (all SantaCruz), anti-5xHis (#34460; Qiagen), rat monoclonal anti-PA (#012-25863; Fujifilm Wako, Japan), and mouse monoclonal anti-1D4 (#40020; Cube Biotech, Monheim, Germany). Rabbit polyclonal antibodies against CLGN, CALR3, PDILT, and OVCH2, mouse monoclonal antibody against tACE, and rat monoclonal antibody against IZUMO1 were obtained as described previously[10,19–21,41]. The following secondary antibodies were also used: peroxidase-conjugated goat anti-rabbit IgG (#111-036-045), goat anti-rat IgG (#112-035-167) and goat anti-mouse IgG (#115-036-062) (all Jackson ImmunoResearch).

### In silico data analysis
GDS403, a microarray dataset for mouse spermatogenesis, was downloaded from the NCBI website. The microarray data were incorporated into Microsoft Excel software and processed according to their expression levels. Single-cell transcriptome data for the murine testis[42] were obtained from https://data.mendeley.com/public-files/datasets/kxd5f8vpt4/files/76bcfd95-5fc5-4256-94cf-a21984138ea5/file_downloaded, and gene expression levels were re-analysed and visualized using Loupe Cell Browser 6.1.0 (10X Genomics).

### Transcript analyses
Total RNA was isolated from mouse tissues using TRIzol reagent (ThermoFisher, #15596026) or an RNeasy mini kit (Qiagen, #74104) for RT-PCR as follows: cDNAs were synthesized from total RNAs using Superscript IV First-Strand Synthesis System (Invitrogen, #18091050) and oligo dT primer according to the manufacturer's instructions. cDNA corresponding to 10 ng of total RNA was used as a template for PCR. The primer sequences and thermal cycling conditions are indicated in Supplementary Table 3. RNA-seq of epididymal transcripts was performed as follows: libraries for sequencing were prepared from isolated RNAs using a TruSeq stranded mRNA sample prep kit (Illumina, #20020594) and sequenced on a NovaSeq6000 (Illumina) using 101 bp single-ended mode. The obtained sequence reads were mapped onto a mouse reference genome (mm10) using TopHat ver. 2.1.1[43]. Reads per kilobase of exon per million mapped reads (RPKM) values were calculated for each gene using Cufflinks ver. 2.2.1[44]. The obtained

RNA-seq data have been deposited in the Gene Expression Omnibus database under the accession code GSE206174.

### Plasmids for protein purification
cDNAs encoding mouse NICOL followed by PA and Rho1D4 epitope tags or 6xHis were cloned into a pCAG vector containing the *CAG* promoter and a rabbit globin poly(A) signal. The plasmid to express C-terminally 6xHis-tagged NELL2 was prepared as described previously[10]. A cDNA encoding the ROS1 extracellular domain followed by 8xHis and Rho1D4 epitope sequence was cloned into the pCAG vector.

### Recombinant protein expression and purification
Human 293-F cells (Gibco, #R79007) were cultured in HE400/AZ chemically defined medium (Gmep, Japan, #HE400AZ-0010) and 6xHis-tagged recombinant NELL2 and NICOL proteins were transiently expressed in 293-F cells using a Gxpress 293 Transfection Kit (Gmep, #GX293-RK-0010) according to manufacturer's instructions. After 5 days of culture, the conditioned medium was harvested by brief centrifugation and the proteins were precipitated by ammonium sulphate at a final concentration of 80%. The precipitate isolated by filtration was then dissolved in 20 mM Tris-HCl pH8.0, 30 mM imidazole, and 1 mM phenylmethylsulfonyl fluoride and loaded onto Ni-NTA agarose resin (Qiagen, #30210). After washing with 20 mM Tris-HCl pH8.0, 30 mM imidazole, and 150 mM NaCl, the bound protein was eluted with 300 mM imidazole pH8.0, desalted using a PD-10 gel filtration column (Cytiva, #17085101) pre-equilibrated with phosphate-buffered saline (PBS), and stored at −80 °C until use. Protein concentration was determined by using a Pierce™ BCA Protein Assay Kit (ThermoFisher, #23225).

### Protein interaction assays
A total of 100 μg of each purified protein was conjugated with 25 μl bed volume of NHS-activated agarose (ThermoFisher, #26200). For NELL2 pulldown, purified recombinant NELL2[10] was applied to protein-conjugated beads, and for ROS1 pulldown, ROS1 ectodomain tagged with 8xHis and Rho1D4 was transiently expressed in 293 F cells, and the transfected cells were lysed with lysis buffer. Cell lysate (1 ml of 1 mg protein/ml) was mixed with 25 μl bed volume of protein-conjugated

agarose beads and incubated overnight at 4 °C with gentle rotation. The beads were then washed three times with 1 ml of lysis buffer, and bound proteins were separated by SDS-PAGE for subsequent immunoblot analysis. Anti-NELL2 antibody and anti-Rho1D4 antibody were used to detect NELL2 and ROS1 ectodomain, respectively.

Surface plasmon resonance assay was carried out by immobilizing purified NELL2 protein onto a series S sensor chip CM5 (Cytiva, #29104988) as a ligand. Purified NICOL protein dissolved in PBS was loaded as an analyte and its association and dissociation kinetics were monitored using Biacore T200 (Cytiva).

*Clgn-Nell2* and *Clgn-Nicol* transgenic mouse testes were homogenized in lysis buffer and protein was extracted. NICOL protein was immunoprecipitated with anti-Rho1D4 monoclonal antibody and Dynabeads protein G (ThermoFisher, #10003D). The immunoprecipitates were analysed by SDS-PAGE followed by immunoblotting and NICOL and NELL2 were detected using anti-PA and anti-NELL2 antibodies, respectively.

### Immunoblot analyses
Tissues were homogenized in lysis buffer (20 mM Tris-HCl pH 7.4, 150 mM NaCl, 1% TritonX-100) containing protease inhibitor cocktail (Nacalai Tesque, Japan, #25955-24) and phosphatase inhibitor cocktail (Nacalai Tesque, #07575-51). The homogenates were gently rotated at 4 °C for 30 min, centrifuged at $12,000 \times g$ at 4 °C for 15 min, and the supernatants were then recovered as crude tissue protein extracts. The protein extracts were separated by SDS-PAGE using e-PAGEL precast gel (Atto, Japan, #E-T/R/D520L) under reducing conditions. Precision Plus Protein Dual Color Standards (Bio-Rad, #1610374) was used as a molecular weight standard. The separated proteins were electrotransferred onto Immobilon-P polyvinylidene difluoride membranes (Merck, #IPVH00010). After blocking with 3% bovine serum albumin (BSA)/TBST, the membranes were incubated overnight with primary antibodies at the indicated dilution (Supplementary Table 4), followed by incubation with horseradish-peroxidase conjugated secondary antibodies. The immunoblot signals were developed using Chemi-Lumi One Super (Nacalai Tesque, #02230) and captured using Amersham ImageQuant 800 (Cytiva).

### Fertility test
Male mice were mated with 2-month-old B6D2F1 WT female mice for several months and females were inspected for the formation of copulatory plugs and delivery every morning. Average litter sizes are presented as the total number of pups born divided by the number of litters for each genotype.

### In vivo sperm migration assay
Male mice carrying the RBGS transgene were mated with WT female mice and the uterus and oviducts were excised 2 h after copulation and placed onto a glass slide. The localization of spermatozoa in the female reproductive tract was visualized by red fluorescence using a fluorescence microscope (BZ-8000; Keyence Corporation, Osaka, Japan).

### In vitro sperm–egg binding assay
The cumulus layer was removed from oocytes by treating with 300 µg/ml hyaluronidase. Cumulus-free oocytes were incubated with spermatozoa isolated from the cauda epididymis and pre-incubated for 2 h in TYH medium at the concentration of $2 \times 10^5$ sperm/ml. After 30 min incubation the oocytes were fixed with 0.25% glutaraldehyde and observed under an inverted microscope (IX70; Olympus, Tokyo, Japan) and spermatozoa bound to the ZP were counted.

### Histology
Tissues were fixed with 4% formaldehyde/PBS overnight, immersed in paraffin, sectioned at 5 µm using a microtome, stained with periodic acid–Schiff or hematoxylin and eosin (HE) and photographed using a system microscope (BX53; Olympus).

### Data representation
Graph representation was performed with GraphPad Prism9.2.0 (MDF).

### Statistical analysis
All experiments were repeated biologically at least three times and similar results were obtained. All statistical analyses were performed using Student's *t*-tests (unpaired two-sided) with Microsoft Excel 2019 (Microsoft). In experiments in which statistical test (two-tailed Student's t-test) was done, the sample size was determined as follows: Cohen's $d = 4$, α error probability = 0.05, and power = 0.8. From these values the sample size for each experimental group was estimated as 3, therefore the sample sizes were $3 \geq n$.

### Reporting summary
Further information on research design is available in the Nature Portfolio Reporting Summary linked to this article.

## Data availability
Source data are provided with this paper. RNA-seq data generated in this study have been deposited in NCBI GEO under accession code GSE206174. Mouse reference genome information is publicly available from NCBI website [https://www.ncbi.nlm.nih.gov/assembly/GCF_000001635.20/]. The main data supporting the results of this study are available from the publisher's website or corresponding authors. Source data are provided with this paper.

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

## Acknowledgements

We thank the NGS core facility of the Genome Information Research Center at the Research Institute for Microbial Diseases of Osaka University for support with RNA sequencing and data analysis and Dr. Julio M. Castaneda for critical reading of the manuscript. This work was supported in part by the Ministry of Education, Culture, Sports, Science and Technology (MEXT)/Japan Society for the Promotion of Science (JSPS) KAKENHI grants (JP21H02487, JP21H00231, and JP21K19263 to D.K. and JP21H05033 and JP19H05750 to M.I.), Japan Science and Technology Agency (JPMJPR2143 to D.K. and JPMJCR21N1 to M.I.), National Institutes of Health (R01HD088412 and P01HD087157 to M.I. and M.M.M.), the Futaba foundation (223074 to D.K.), the Japan Foundation for Applied Enzymology (2022-12 to D.K.), and the Bill & Melinda Gates Foundation (Grant INV-001902 to M.I. and M.M.M.). Under the grant conditions of the Foundation, a Creative Commons Attribution 4.0 Generic License has already been assigned to the author accepted manuscript version that might arise from this submission.

## Author contributions

D.K., K.S., M.M.M., D.H.W., and M.I. designed the experiments. D.K., K.S., M.C., M.K., S.O., C.E., and T.N. performed the experiments. T.E. assisted the interpretation of testis histological analysis. D.K., K.S., M.M.M., D.H.W., and M.I. wrote and revised the manuscript. The first authors D.K. and K.S. contributed equally to this research project. All authors read, corrected, and approved the manuscript.

## Competing interests

The authors declare no competing interests.
