## [Peer Review File · Nature Communications]

A small secreted protein NICOL regulates lumicrine-mediated sperm maturation and male fertilityREVIEWER COMMENTS

Reviewer #1 (Remarks to the Author):

In this manuscript the authors identify a novel epididymal-secreted protein, which they name NICOL, as a mediator of male fertility using a mouse model. This is performed using appropriate methodology involving gene editing and rescue. and, although several other proteins have been identified in the same (or similar) manner (and also considered as targets for male contraception) this data is of particular interest given how it addresses issues of male fertility and sperm competence beyond the testis. In this case NICOL seems relevant for the sperm to reach the oviduct and bind ZP.

However, the key of the mechanism proposed was previous 2020 paper, featuring the protein NICOL binds to (and is a co-factor of) NELL2. In that regard the novelty is clearly lower and the increase in knowledge incremental from that immediate standpoint, despite the quality of the data.

I have a few comments and notes for the authors:

-NICOL seems widely expressed, including in the brain, but the the KO mice (assuming they are not tissue specific, it was not immediately clear- if they were disregard) seem to have no other phenotype beyond male infertility. Was behavior looked into at all, for example? What I mean are phenotypes beyond fertility, I know this is not the immediate point, but I think it would be important to discuss.

-basic sperm parameters (number, motility patterns and morphology) would be important in order to determine the true function of NICOL.

-Besides fertilization, if early embryo development normal when it does occur for KO sperm (in the absence of ZP)? This would lead to answers regarding other issues of sperm competence.

-Testis IHC images focusing on NICOL and NELL2 co-localization (besides RNAseq) would be important to validate the data.

-It might not be significant, but in some cases (Fig 6, for example) IZUMO levels, which are used as an internal control, seem lower in the NICOL KO. Do the authors have any information on other relevant protein levels in their NICOL KO model?

-Notably, are NELL2 or ROS1 levels affected in the NICOL KO? I'm sorry if that information was available and I missed it.

Reviewer #2 (Remarks to the Author):

This manuscript examines the molecular mechanism of lumicrine signaling in the epididymis. The work presented is an extension of work done by the investigative team to delineate the signals from the testis that trigger epididymal differentiation and sperm maturation. It has long been known in the epididymal field that sperm maturation is under the control of luminal factors released by the testis and that maintain the overall architecture and function of the first part of the epididymis known as the initial segment which plays critical roles in initiating sperm maturation. These investigators have made significant progress towards identifying the players in the lumicrine signaling pathway and in the present report identify another member, Nicol, that is essential for lumicrine signaling. Therefore the work is certainly significant for the epididymal field as not only does it inform on how sperm maturation is controlled but also provides potential targets for the development of male contraceptives. To strengthen their results, this reviewer has the following suggestions:

1) Fig 1. The authors state that Nicol ablation did not affect the morphology of epididymal spermatozoa yet show in their representative image sperm that is kinked in the Nicol KO but which is absent from wildtype. As this characteristic is often indicative of epididymal dysfunction, the authors need to quantify whether these cells are more prevalent in the KO or are also present in WT samples. (maybe a table showing the distribution of normal vs abnormal morphologies?)

2) Fig3. The authors show that normal ADAM3 processing does not occur in the Nicol KO. It would be helpful if full immunoblot images were provided to demonstrate that other sized ADAM3 isoforms were

not present, suggesting aberrant processing.

3) The authors begin their results emphasizing the initial segment (IS) region of the epididymis as the target for lumicrine signaling (which has long been known) and demonstrate that Nicol KO IS does not differentiate properly (as previously demonstrated by others for the ROS KO). In subsequent figures the authors jump to the caput region which is distal to the IS and which previous data by many others suggest is under a different regulation than the IS. It is in this region that the authors focus their studies on showing changes of ERK and OVCH2 etc in the Nicol KO. Similarly, the RNA seq analysis showing genes regulated by lumicrine signaling is also from the caput. There is no discussion of this transition as to why they are no longer studying the IS. This includes in their working model of Nicol lumicrine signaling which I found lacking as it only emphasizes NELL2/NICOL interactions rather than a bigger picture of the communication between the testis and epididymis. Do the authors think the first target is ROS in the IS with signals then sent from this region (Nicol?) to the caput or do they have evidence their molecules are similarly regulating both the IS and the caput? The RTPCR studies in Fig1 only show Nicol mRNA in caput, corpus, and cauda. Why was IS excluded from this analysis or was it included as part of the caput? Is Nicol expressed in the IS region? Overall a more complete working model would be helpful (even if all the gaps are not yet filled in.. these could be indicated with ?).

4) The authors also state in their results line 113-115 than secretions from the pachytene spermatocytes constitute the testicular luminal fluid. If so, a reference showing data to support this statement is needed since testicular fluid components are thought to be mostly from Sertoli cells not spermatocytes. The authors make no mention of work published decades ago suggesting the lumicrine factors that regulate the IS are associated with sperm (which would support their contention). The source of Nicol in testicular fluid is not demonstrated and the authors have not ruled out that Nicol is associated with the sperm surface. These caveats need to be mentioned.

POINT-BY-POINT RESPONSES TO REVIEWERS' COMMENTS

Reviewer #1 (Remarks to the Author):

In this manuscript the authors identify a novel epididymal-secreted protein, which they name NICOL, as a mediator of male fertility using a mouse model. This is performed using appropriate methodology involving gene editing and rescue. and, although several other proteins have been identified in the same (or similar) manner (and also considered as targets for male contraception) this data is of particular interest given how it addresses issues of male fertility and sperm competence beyond the testis. In this case NICOL seems relevant for the sperm to reach the oviduct and bind ZP.

However, the key of the mechanism proposed was previous 2020 paper, featuring the protein NICOL binds to (and is a co-factor of) NELL2. In that regard the novelty is clearly lower and the increase in knowledge incremental from that immediate standpoint, despite the quality of the data.

I have a few comments and notes for the authors:

-NICOL seems widely expressed, including in the brain, but the KO mice (assuming they are not tissue specific, it was not immediately clear- if they were disregard) seem to have no other phenotype beyond male infertility. Was behavior looked into at all, for example? What I mean are phenotypes beyond fertility, I know this is not the immediate point, but I think it would be important to discuss.

For behavioral aspects, the mating behavior was not affected by *Nicol* ablation, as described (page 3 lines 53 – 56 in the initial manuscript, which corresponds to page 3 lines 58 – 60 in the revised manuscript) because the copulatory plugs were successfully formed by *Nicol*^{-/-} males. Besides, at least the feeding behavior appeared unaffected by *Nicol* ablation. We also measured the body weight, which is now included in the Extended Data Figure 1d, to evaluate feeding and growth aspects and there was no significant difference between WT and *Nicol*^{-/-} males. About behaviors beyond mating and feeding, however, we would like to leave these issues to be addressed in the future because neuroscience and behavioral science are outside our expertise. These observations are described on page 2 line 48 – page 3 line 50 and page 3 line 58 – 60 of the revised manuscript.

-basic sperm parameters (number, motility patterns and morphology) would be important in order to determine the true function of NICOL.

We have counted the total number of spermatozoa and evaluated the morphology (i.e., straight, bent, and hairpin) of spermatozoa as shown in Extended Data Figure 6a and b in the revised manuscript. We have also included the computer-assisted sperm analysis (CASA) results of spermatozoa, which are shown in Extended Data Figure 6c in the revised manuscript. The number of cauda spermatozoa of *Nicol*^{-/-} mice were comparable to that of control animals but they were often

bent. The motility of *Nicol*^{-/-} spermatozoa was slightly worse compared with control ones. The observed sperm abnormalities of *Nicol*-null mice are the phenocopy of *Ros1*^{-/-} spermatozoa (Yeung et al., *Biol. Reprod.* 1999;61:1062–9.), as the epididymal luminal environment necessary for sperm maturation was ablated by defective initial segment differentiation in both mutant mouse lines. These observations are also briefly described on page 5 lines 110– 114 of the revised manuscript.

-Besides fertilization, if early embryo development normal when it does occur for KO sperm (in the absence of ZP)? This would lead to answers regarding other issues of sperm competence.

We have obtained fertilized eggs by inseminating cumulus-intact oocytes (we did not use zona-free oocytes to avoid polyspermic fertilization) with *Nicol*-null spermatozoa and found that they successfully developed into blastocysts (Figure 2n-p in the revised manuscript). These results indicate the competence of *Nicol*-null spermatozoa as gametes. These observations are described on page 3 lines 69 – 70 of the revised manuscript.

-Testis IHC images focusing on NICOL and NELL2 co-localization (besides RNAseq) would be important to validate the data.

We performed immunostaining of testis sections with anti-NICOL antibody was examined but were unsuccessful (as shown in the Figure only for editor and reviewers), probably because of low expression of target protein, washing out of soluble proteins during immunostaining, and/or incompatibility of available antibody for immunostaining. Instead, we have analyzed single-cell RNA-sequencing data in detail; As shown in Figure 5d and described on page 5 lines 124 – 125 of the revised manuscript, *Nell2*-expressing cells also co-express *Nicol*.

-It might not be significant, but in some cases (Fig 6, for example) IZUMO levels, which are used as an internal control, seem lower in the NICOL KO. Do the authors have any information on other relevant protein levels in their NICOL KO model?

Since *Nicol*-null sperm successfully fertilize with ZP-free oocytes, such a change in IZUMO1 expression level does not critically affect sperm-oocyte fusion. We added immunoblots against sperm surface tACE and mitochondrial TOM20 as additional internal controls, as shown in the revised Figure 6f. Since altered luminal environment of *Nicol*^{-/-} epididymis can affect sperm fragility, the recovery yield of sperm proteins during experimental preparation from cauda epididymis may be a little worse.

-Notably, are NELL2 or ROS1 levels affected in the NICOL KO? I'm sorry if that information was available and I missed it.

Our initial manuscript had already included the RT-PCR results of *Nell2* and *Ros1* of *Nicol*^{-/-}

mice, which correspond to Figure 4k and page 5 lines 116 – 120 in the revised manuscript; the expression of neither testicular *Nell2* nor epididymal *Ros1* was compromised by *Nicol* ablation.

Reviewer #2 (Remarks to the Author):

This manuscript examines the molecular mechanism of lumicrine signaling in the epididymis. The work presented is an extension of work done by the investigative team to delineate the signals from the testis that trigger epididymal differentiation and sperm maturation. It has long been known in the epididymal field that sperm maturation is under the control of luminal factors released by the testis and that maintain the overall architecture and function of the first part of the epididymis known as the initial segment which plays critical roles in initiating sperm maturation. These investigators have made significant progress towards identifying the players in the lumicrine signaling pathway and in the present report identify another member, *Nicol*, that is essential for lumicrine signaling. Therefore the work is certainly significant for the epididymal field as not only does it inform on how sperm maturation is controlled but also provides potential targets for the development of male contraceptives. To strengthen their results, this reviewer has the following suggestions:

1) Fig 1. The authors state that *Nicol* ablation did not affect the morphology of epididymal spermatozoa yet show in their representative image sperm that is kinked in the *Nicol* KO but which is absent from wildtype. As this characteristic is often indicative of epididymal dysfunction, the authors need to quantify whether these cells are more prevalent in the KO or are also present in WT samples. (maybe a table showing the distribution of normal vs abnormal morphologies?)

We have evaluated the total number and the morphology of *Nicol*-null cauda spermatozoa as shown in Extended Data Figure 6a and b and on page 5 line 110 – 114 of the revised manuscript.

2) Fig3. The authors show that normal ADAM3 processing does not occur in the *Nicol* KO. It would be helpful if full immunoblot images were provided to demonstrate that other sized ADAM3 isoforms were not present, suggesting aberrant processing.

A new anti-ADAM3 immunoblot image covering a whole molecular weight range is included in Extended Data Figure 2 and on page 4 lines 79 – 80 in the revised manuscript.

3) The authors begin their results emphasizing the initial segment (IS) region of the epididymis as the target for lumicrine signaling (which has long been known) and demonstrate that *Nicol* KO IS does not differentiate properly (as previously demonstrated by others for the ROS KO). In subsequent figures the authors jump to the caput region which is distal to the IS and which previous data by many

others suggest is under a different regulation than the IS. It is in this region that the authors focus their studies on showing changes of ERK and OVCH2 etc in the Nicol KO. Similarly, the RNA seq analysis showing genes regulated by lumicrine signaling is also from the caput. There is no discussion of this transition as to why they are no longer studying the IS. This includes in their working model of Nicol lumicrine signaling which I found lacking as it only emphasizes NELL2/NICOL interactions rather than a bigger picture of the communication between the testis and epididymis. Do the authors think the first target is ROS in the IS with signals then sent from this region (Nicol?) to the caput or do they have evidence their molecules are similarly regulating both the IS and the caput? The RTPCR studies in Fig1 only show Nicol mRNA in caput, corpus, and cauda. Why was IS excluded from this analysis or was it included as part of the caput? Is Nicol expressed in the IS region? Overall a more complete working model would be helpful (even if all the gaps are not yet filled in.. these could be indicated with ?).

While it is easy to identify the IS on tissue sections, it is difficult to dissect only the IS for protein and RNA extraction. Especially, it is extremely difficult to visually dissect only the region corresponding to the IS from *Nicol*^{-/-} epididymis in which IS differentiation is not observed. Therefore, in this study, IS and caput were dissected together when IS needs to be isolated for the comparative study (the dissected tissue was described as “caput” in the first manuscript). However, as pointed out by reviewer #2, this description has been corrected to be more explicit in the revised manuscript; when IS and caput were dissected together to apply to subsequent analyzes (i.e., Figures 1b, 3d, 4i, 4j, 4k, 6b, 6e, Extended Data Figure 2, and Extended Data Figure 5), the description “caput” was changed to “IS-caput.” This is also described in Methods: page 9 lines 228 – 231 of the revised manuscript.

Nicol is also expressed in the IS; to examine *Nicol* expression in the IS, we also added RT-PCR results of the initial to the fifth segments of epididymis, which is included in the revised Figure 1b.

A new working model including the complete process of NICOL secretion, complex formation with NELL2, luminal transport, and action on ROS1 expressed in IS epithelial cells is shown in the new Figure 7 and described on page 7 line 178 – page 8 line 187 of the revised manuscript.

4) The authors also state in their results line 113-115 than secretions from the pachytene spermatocytes constitute the testicular luminal fluid. If so, a reference showing data to support this statement is needed since testicular fluid components are thought to be mostly from Sertoli cells not spermatocytes. The authors make no mention of work published decades ago suggesting the lumicrine factors that regulate the IS are associated with sperm (which would support their contention). The source of Nicol in testicular fluid is not demonstrated and the authors have not ruled out that Nicol is associated with the sperm surface. These caveats need to be mentioned.

As pointed out by reviewer #2, testicular luminal fluid (seminiferous tubular fluid) is generated

by Sertoli cells (Review by Rato et al., *J. Membr. Biol.* 2010;236:215-24); pumping out of Na⁺ by Na⁺/K⁺ ATPase on the apical surface of Sertoli cells and Cl⁻ efflux into apical space by CFTR in conjunction with Na⁺ movement generate osmotic gradient across the apical membrane of Sertoli cells. To eliminate this osmotic gradient, water molecules move toward the apical lumen through aquaporin and this water movement generates a volume of luminal fluid. About protein secretions existing in the luminal fluid such as NICOL, any cells facing the apical lumen such as Sertoli cells and spermatocytes can theoretically contribute through exocytosis. Collectively, the contribution of NICOL (and NELL2) as luminal fluid solute from spermatocytes is reasonable. This is briefly described on page 7 line 179 – page 8 line 182 of the revised manuscript.

As also pointed out by reviewer #2, lumicrine factor transport by sperm was previously suggested (Garrett et al., *Mol. Endocrinol.* 1990;4:108-18.). If it is so, lumicrine signaling never occurs before completion of spermatogenesis. However, as shown in Extended Data Figure 2b, in which IS differentiation is histologically apparent in 4-week-old epididymis and precedes completion of spermatogenesis. Besides, it is reported that the IS response to lumicrine factor starts to occur on postnatal day 19 (Xu et al., *Biol. Reprod.* 2016;95:15). Collectively, these findings do not immediately exclude the contribution of sperm as lumicrine factor carrier but strongly support the sperm-independent lumicrine factor transport. This is briefly described on page 8 lines 182 – 185 of the revised manuscript.

REVIEWERS' COMMENTS

Reviewer #1 (Remarks to the Author):

The authors have adequately addressed my concerns in this revised version. I have no further comments.

Reviewer #2 (Remarks to the Author):

The authors have strengthened their revised manuscript by adding sperm number, motility, and morphology data, including % bent sperm, from the NICOL KO mice. They have also clarified in the figures that samples were prepared from initial segment and caput combined. Below are comments for the authors.

It is unclear what they are showing in the RTPCR data in Fig 1b. A new panel has been added which is indicated as IS-caput but the legend states initial segment to the fifth segment of the epididymis (1-5). Do the authors mean they sectioned the IS-caput into 5 distinct segments?

It seems from the RNA data in Fig 5b that NICOL is also expressed in Sertoli cells. Please clarify. If so the presence of NICOL in luminal fluid could represent contributions from these cells in addition to other cell populations. I raise this point since the authors continue to state in the results that secretions from pachytene spermatocytes constitute the testicular luminal fluid which, as far as I know, has never been examined/demonstrated. The Sertoli cells are thought to be the primary source. A reference supporting their statement was requested but was not provided in their revision. While it might be possible, to make this statement as fact is misleading. In addition, the authors continue this line of thought in their model. Indeed, pachytene spermatocytes do not contact the adluminal compartment, spermatids do. I realize that their rescue experiments indicate germ cells as the critical cells, but the authors need to modify their statements to clearly indicate what is known and what, for now, is pure speculation (e.g our data suggest spermatids secrete NICOL but further experiments are needed to establish this).

POINT-BY-POINT RESPONSES TO REVIEWERS' COMMENTS

Reviewer #2 (Remarks to the Author):

The authors have strengthened their revised manuscript by adding sperm number, motility, and morphology data, including % bent sperm, from the NICOL KO mice. They have also clarified in the figures that samples were prepared from initial segment and caput combined. Below are comments for the authors.

It is unclear what they are showing in the RTPCR data in Fig 1b. A new panel has been added which is indicated as IS-caput but the legend states initial segment to the fifth segment of the epididymis (1-5). Do the authors mean they sectioned the IS-caput into 5 distinct segments?

We are sorry for this confusion. In the revised Figure 1b the initial, 2nd, 3rd, 4th, and 5th segments are indicated.

It seems from the RNA data in Fig 5b that NICOL is also expressed in Sertoli cells. Please clarify.

Yes, you are correct. The expression of *Nicol* in germ cells and Sertoli cells is now described on page 6 lines 142-143 of the revised manuscript as follows: “*Nicol* is expressed in both germ cells and Sertoli cells (Fig. 5b).”

If so the presence of NICOL in luminal fluid could represent contributions from these cells in addition to other cell populations. I raise this point since the authors continue to state in the results that secretions from pachytene spermatocytes constitute the testicular luminal fluid which, as far as I know, has never been examined/demonstrated. The Sertoli cells are thought to be the primary source. A reference supporting their statement was requested but was not provided in their revision. While it might be possible, to make this statement as fact is misleading.

Thank you for bringing up this point. We agree that the seminiferous luminal fluid is generated by Sertoli cells, as described in page 9 lines 199-200 “The testicular seminiferous luminal fluid is generated by the function of ion transporters and aquaporins in Sertoli cells” and in the previous PBP response. Based on the high level of expression of NICOL and NELL2 in spermatocytes, the spermatocytes are the source of these proteins (lumicrine factors) that are secreted into the luminal fluid. Hopefully we have explained this more clearly.

In addition, the authors continue this line of thought in their model. Indeed, pachytene spermatocytes do not contact the adluminal compartment, spermatids do.

All cells located inside the blood-testis barrier, which is emphasized in the revised Figure 5b, a tight

junction, can face apical space or seminiferous luminal space. This is a very reasonable conclusion based on the widely accepted consensus of cell biology; it can also be found in many review articles schematically explaining the structure of the seminiferous tubule.

Right, An example to explain pachytene spermatocytes as well as spermatids are facing the luminal fluid, from *Front. Endocrinol.* 2022;13:897062. doi: 10.3389/fendo.2022.897062.

Figure from: Hofmann M-C and McBeath E (2022) Sertoli Cell-Germ Cell Interactions Within the Niche: Paracrine and Juxtacrine Molecular Communications. *Front. Endocrinol.* 13:897062. doi: 10.3389/fendo.2022.897062

I realize that their rescue experiments indicate germ cells as the critical cells, but the authors need to modify their statements to clearly indicate what is known and what, for now, is pure speculation (e.g our data suggest spermatids secrete NICOL but further experiments are needed to establish this). We described “The NELL2 and NICOL proteins secreted into seminiferous luminal fluid form a complex and reach the epididymal IS by the luminal flow. Spermatocytes seems to play a critical role in lumicrine factor secretion as they express both *Nell2* and *Nicol* at the highest level in testicular cells.” in the legend of Figure 7b, which represents a possible model of lumicrine factor secretion and transport, in page 28 lines 652-656 of the revised manuscript. We do not specify spermatocytes as the sole source of lumicrine factor and do not exclude spermatid but suggest their possible major contribution upon gene expression level. It is generally reasonable to consider the contribution of cell population expressing highest levels of genes of interest.